# Modelling the impact of migrants on the success of the HIV care and treatment program in Botswana

Tafireyi Marukutira[1,2], Nick Scott[1], Sherrie L. Kelly[1], Charles Birungi[3,4], Joseph M. Makhema[5], Suzanne Crowe[1,2], Mark Stoove[1,2], Margaret Hellard[1,2] *

1 Burnet Institute, Melbourne, Australia, 2 Monash University, Melbourne, Australia, 3 UNAIDS, Gaborone, Botswana, 4 University College London, London, England, United Kingdom, 5 Botswana Harvard Partnership, Gaborone, Botswana

* margaret.hellard@burnet.edu.au

## Abstract

### Introduction

Botswana offers publicly financed HIV treatment to citizens, but not migrants, who comprised about 7% of the population in 2016. However, HIV incidence is not declining in proportion to Botswana's HIV response. In 2018, Botswana had 86% of citizens living with HIV diagnosed, 95% of people diagnosed on treatment, and 95% viral suppression among those on treatment. We hypothesised that continued exclusion of migrants is hampering reduction of HIV incidence in Botswana. Hence, we modelled the impact of including migrants in Botswana's HIV response on achieving 90-90-90 and 95-95-95 Fast-Track targets by 2020 and 2030, respectively.

### Methods

The Optima HIV model, with demographic, epidemiological, and behavioural inputs, was applied to citizens of and migrants to Botswana. Projections of new HIV infections and HIV-related deaths were compared for three scenarios to the end of 2030: (1) continued status quo for HIV testing and treatment coverage, and maintenance of levels of linkage to care, loss to follow-up, and viral suppression among citizens and migrants (baseline); (2) with scaled-up budget, optimised to achieve 90-90-90 and 95-95-95 Fast-Track targets by 2020 and 2030, respectively, for citizens only; and (3) scaled-up optimised budget to achieve these targets for both citizens and migrants.

### Results

A baseline of 172,000 new HIV infections and 8,400 HIV-related deaths was projected over 2020–2030. Scaling up to achieve targets among citizens only averted an estimated 48,000 infections and 1,700 deaths. Achieving targets for both citizens and migrants averted 16,000 (34%) more infections and 442 (26%) more deaths. Scaling up for both populations reduced numbers of new HIV infections and deaths by 44% and 39% respectively compared

**Data Availability Statement:** All relevant data are within the manuscript and its Supporting Information files.

**Funding:** The authors received no specific funding for this work.

**Competing interests:** The authors have declared that no competing interests exist.

with 2010 levels. Treating migrants when scaling up in both populations was estimated to cost USD 74 million over 2020–2030.

## Conclusions

Providing HIV services to migrants in Botswana could lead to further reductions in HIV incidence and deaths. However, even with an increased, optimised budget that achieves 95-95-95 targets for both citizens and migrants by 2030, the 90% incidence reduction target for 2020 will be missed. Further efficiencies and innovations will be needed to meet HIV targets in Botswana.

## Introduction

Reaching the UNAIDS 90-90-90 Fast-Track targets by 2020 and further scaling up to 95-95-95 by 2030 is hypothesized to lead to a 90% reduction in HIV incidence and HIV-related mortality from 2010 levels [1]. Additional epidemiological transition metrics for tracking the HIV epidemic, such as the incidence:prevalence and the incidence:mortality ratios have also been proposed [2]. One of the major assumptions for achieving these reductions in the Fast-Track targets is that scale-up of effective prevention strategies, testing, treatment and viral load monitoring is applied to all population groups. However, some country programs often exclude key population groups vulnerable to HIV infection, including migrants.

With more than 258 million global international migrants in 2017, migration continues to be a key consideration in public health programs [3]. In sub-Saharan Africa, the region most affected by the HIV epidemic, migration and public health is even more pertinent. Migrants often face barriers to healthcare access and are often excluded from mainstream national programs [4, 5]. The role of migration in the HIV epidemic is complex. While migration itself is not a risk factor for HIV infection, human mobility places people in situations that increase their risk of acquiring HIV and impact on them seeking timely care if infected (e.g., irregular migration status and lack of migrant-inclusive health policies, which may prevent early healthcare seeking) with the potential risks occurring both pre- and post-migration [6]. However, research findings can be mixed; some research has demonstrated that migrants are at increased risk of post-migration HIV acquisition while other studies showed no positive association between intensity of migration measured at country level and peak HIV prevalence [7–10]. The risk of post-migration HIV acquisition has not been clearly elucidated. Among migrants living with HIV, the proportion that acquired HIV post-migration has been reported to be as low as 2% and as high as 71% and can depend on the prevalence within their country/region of origin [8, 11].

HIV prevalence is high in sub-Saharan Africa, including in Botswana, but unlike some countries in the region Botswana has a strong economy so attracts many migrants [12]. In 2015, it was estimated that Botswana had 161,000 documented migrants, approximately 7% of its total population [13]. Migrants to Botswana are mainly from within the region, especially Zimbabwe. Adult HIV prevalence is estimated at 20% among migrants, compared with 22.8% nationally [12, 14]. Botswana has a free nationally funded HIV care and treatment program that covers all citizens; the system is considered to be highly successful, with Botswana being reported to be close to reaching the 90-90-90 targets. The government of Botswana funds at least 60% of its HIV response budget, which is expected to rise to USD339 million in 2030 [15, 16]. In 2018, Botswana had 86% of citizens living with HIV diagnosed, 95% of people diagnosed on antiretroviral therapy (ART), and 95% viral suppression among those on ART (i.e.

86-95-95) [17]. However, HIV incidence remains a concern, with minimal incidence reductions observed between 2010 and 2018 [6, 18–20]. Importantly, migrants living with HIV are excluded from the free national HIV care and treatment program and many cannot afford to pay for their own HIV care. The implications of excluding migrants from mainstream HIV programs in Botswana is unknown, but may reduce the feasibility of achieving a 90% reduction in HIV incidence even if the Fast-Track targets are met among citizens.

We hypothesized that continued exclusion of migrants is slowing progress in reducing HIV incidence in Botswana. To test this hypothesis, we modelled the impact of including migrants in Botswana's HIV response on achieving 90-90-90 and 95-95-95 Fast-Track targets by 2020 and 2030, respectively.

## Methods

We applied the Optima HIV model, a dynamic, population-based HIV model to test our hypothesis (described in detail elsewhere [21, 22]). The Optima model tracks the entire population of people living with HIV (PLHIV) in a country or region between health states: from infection to diagnosis, linkage to care, ART initiation, viral suppression, and death; and across CD4+ count stages (acute HIV infection, >500, 350–500, 200–350, 50–200, and <50 cells/μL). The overall population is partitioned by population group and by HIV health state. The Optima HIV model is used for modeling HIV epidemics and the impact of interventions to address policy and program challenges. The model can project countries' progress towards the UNAIDS 90-90-90 targets, including HIV incidence and mortality.

Two populations were considered in the model: citizens of Botswana and migrants in Botswana. Migrants were defined as individuals who emigrated to Botswana. The model requires a range of demographic, epidemiological, and behavioral data inputs broadly related and these include population size; prevalence values for HIV, sexually transmitted infections (STIs), and tuberculosis (TB); risk behavior data (e.g., condom use); and biological constants (e.g., disease progression). HIV transmission was determined by the number and type of risk events (either within individuals' population groups or through interaction with other population groups) and the infection probability of each event. Botswana's HIV epidemic is driven predominantly by heterosexual sexual transmission, so the probability of people in the model becoming infected was determined by: 1) HIV prevalence (weighted by viral load) in partner populations; 2) average number of casual, regular, and commercial homosexual and heterosexual acts per person per year; 3) proportion of sexual acts in which condoms are used; 4) proportion of men who are circumcised; 5) prevalence of STIs; 6) proportion of acts that are covered by pre-exposure prophylaxis (PrEP) and post-exposure prophylaxis (PEP); 7) number of sexual partners; and 8) efficacy values for condoms, male circumcision, PEP, PrEP, and ART (suppressive and non-suppressive) in preventing HIV transmission (see supporting information, S1 and S2 Tables).

### Data sources

Epidemiological, behavioral, and programmatic data, and estimates to inform the model, were collated from literature reviews and country program data between 2010 and 2018, and were validated in consultation with key country stakeholders. Two populations were considered in this analysis: citizens and documented migrants aged 15–65 years, each disaggregated by sex. Model projections were generated to 2030 inclusive. Table 1 lists the key data inputs to inform this modeling analysis.

**Table 1. Key model inputs.**

| Indicator | Value [Reference] | |
| --- | --- | --- |
| | **Citizens** | **Migrants** |
| Population size[£] | 1,604,400 (2014) [23] | 112,195 (2016) [13] |
| Estimated HIV prevalence (%)[£] | 19.5% (2016) [12, 19] | 15% (2017) [14] |
| HIV testing | 31–55% tested (2017) [24] | 35%* |
| Number of HIV diagnoses per year | 2010: 13,000; 2013: 9,100; 2016: 10,000; 2017: 14,000 [24] | 1,300* |
| PLHIV aware of their status (%) | 86% (83–89%) [12, 20, 24] | 25% [14] |
| Average time taken to be linked to care (years) | 0.3* | 0.9* |
| Average time taken to be linked to care for people with CD4<200 (years) | 0.5* | 0.5* |
| Total number of people on treatment | 380,000 [12] | 18,600* |
| PLHIV in care on treatment (%) | 84–90% [12, 20] | 76% [14] |
| Treatment failure rate (%) | 7%* | 25%* |
| Percentage of people in care who are lost to follow-up per year (%/year) | 5%* | 15%* |
| People on ART with viral suppression (%) | 95% [12, 20] | 82% [26] |
| Pregnant women on PMTCT (%) | 2015: 92%; 2016: 95% [24] | 60%* |
| Number of women on PMTCT (Option B/B+) | 120,000 [24] | 500* |
| Average number of sex acts with regular partners per person per year | 54* | 54* |
| Percentage of people who used a condom at last act with regular partners | 82% [24] | 82%* |
| Percentage of males who have been circumcised | 45% [19] | 30%* |
| Number of HIV-related deaths | 14,000 in 2005 and 3,900 in 2016 [12, 24, 25] | 1,500*; 8% of PLHIV on ART [26] |
| Efficacy of suppressive ART | 1.0 | 1.0 |

Efficacy of other prevention interventions: N/A: interventions are implemented as relative risks, and coverage was assumed to remain constant for projected scenario.

[£]A factor was considered to represent the targeted age group (15–65 years), see S1 Table for additional justifications;

*Assumption for 2017 inputs as data was not available;

ART: antiretroviral therapy; PLHIV: people living with HIV; PMTCT: prevention of mother-to-child transmission.

## Model calibration

We initialized the model in 2010 and produced projections from 2020 to 2030 inclusive. We fitted the model to key data points: population size, HIV prevalence, PLHIV on treatment, HIV-related deaths, and new HIV diagnoses. To get the best fit, we adjusted the following parameters: force of infection (which depends on the probability of infection, diagnosis rates and mixing of population groups), inhomogeneity of the population groups, and efficacy of suppressive ART [27]. This was initially done using the auto-calibrate feature, which uses an optimisation to vary the calibration parameters simultaneously to minimise the model error from the data points. The auto-calibrated parameters can then be manually adjusted in the web-interface to fine-tune the model fit.

## Analytical approach

Projections of new HIV infections and HIV-related deaths were compared for three scenarios to the end of 2030: (1) status quo for HIV testing and treatment coverage, and levels for linkage

to care, loss to follow-up, and viral suppression maintained among citizens and migrants as a baseline scenario; (2) with scaled-up budget under optimized allocation to achieve 90-90-90 and 95-95-95 Fast-Track targets by 2020 and 2030, respectively, for citizens only; and (3) scaled-up optimized budget to achieve these targets for both citizens and migrants. Mixing patterns and the coverage of other prevention programs were assumed to remain constant.

For each scenario, we used the model to project the cumulative number of new HIV infections and HIV-related deaths between 2020 (base year) and 2030 inclusive, and the percentage change in annual new HIV infections, HIV incidence and HIV-related deaths by 2030 inclusive compared with 2020. These outcomes were obtained for the total Botswana population and separately for citizens and migrants. We also compared the projected new infections in 2030 with the published estimates for 2010 [28]. Assuming a conservative cost of USD300 per person per year for treating an individual with ART in Botswana, we estimated the cost required to treat migrants per year when scaling up in both populations [29]. Cost of HIV treatment includes drug costs, laboratory work, and service delivery.

## Results

For the baseline scenario, 172,000 cumulative new HIV infections were projected to occur from 2020 to 2030. For the baseline scenario, are projected to increase from 13,000 in 2020 to 18,800 in 2030. Scaling up to achieve 90-90-90 and 95-95-95 Fast-Track targets among citizens averted an estimated 48,000 infections, and achieving targets for both citizens and migrants over this period averted an additional 16,000 infections (34% more) (Fig 1). Thus, scaling up in citizens was projected to result in 124,000 cumulative new infections in 2020–30 (inclusive), while scaling up in citizens and migrants was projected to result in 108,000 new infections. Scaling up in both populations would reduce HIV incidence from 0.64 per 100 person-years (12,500 new HIV infections) in 2020 to 0.23 per 100 person-years (7,300 new HIV infections) in 2030.

Compared with 2020 levels, new HIV infections in the total Botswana population in 2030 were projected to be 31% higher (29% higher in migrants and 31% higher in citizens) in the

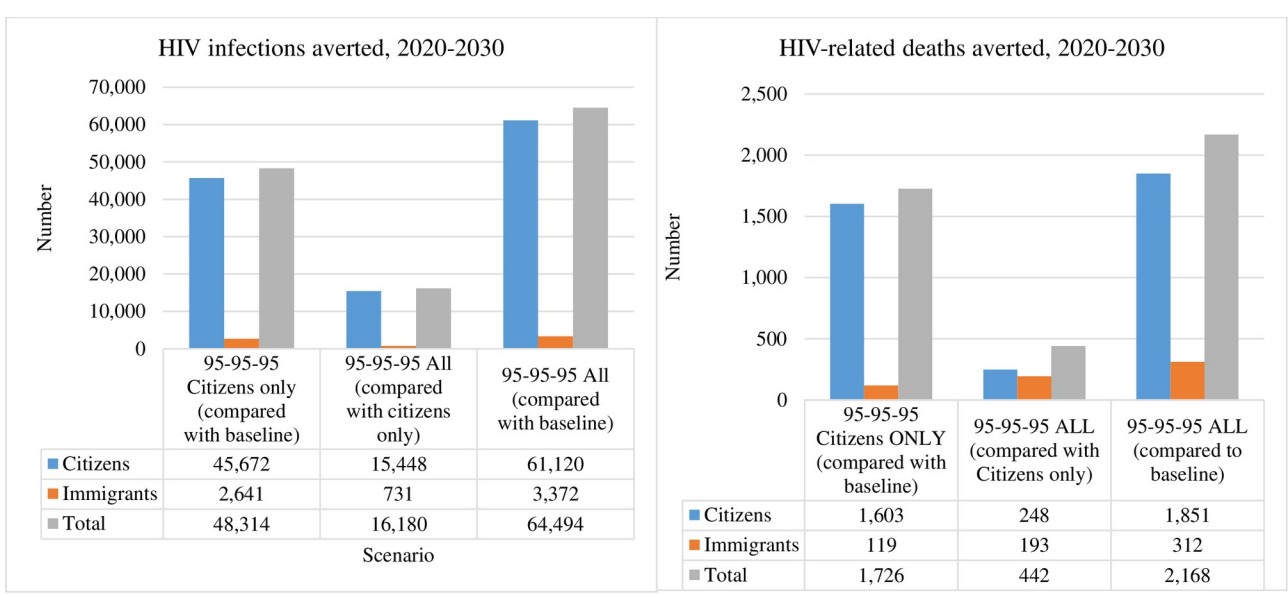

**Fig 1. HIV infections and HIV-related deaths averted.** HIV infections and HIV-related deaths that could be averted if 2030 95-95-95 targets are achieved for citizens only or both citizens and migrants (all) compared to baseline scenario, 2020–30

baseline scenario, 23% lower (23% lower in both citizen and migrants) when scaling up in citizens only, and 42% lower (41% lower in migrants and 42% lower in citizens) when scaling up in both population groups (Table 2). Relative to the estimated 13,000 new HIV infections in Botswana in 2010, scaling up to Fast Track 95-95-95 targets by 2030 in both population groups was projected to result in a 44% reduction in new HIV infections in 2030 (7,300 projected in 2030), well short of the 90% reduction in HIV incidence target [28]. Fig 2 presents new HIV infection trends over time for each scenario and population group.

The baseline scenario was projected to result in 8,400 HIV-related deaths between 2020 and 2030, compared with 6,700 and 6,200 when scaling up in citizens only or in both population groups respectively (1,700 and 2,200 deaths averted, respectively) [Table 2]. This translates to 26% more deaths being averted when scaling up in both citizens and migrants. Compared with 2020 levels, HIV-related mortality in the total Botswana population in 2030 was projected to be 15% higher in the baseline scenario (13% higher in migrants and 15% higher in citizens), 28% lower (no change for migrants and 31% lower in citizens) when scaling up in citizens only, and 39% lower (58% lower in migrants and 37% lower in citizens) when scaling up in both population groups (Fig 3).

Treating migrants living with HIV was estimated to cost USD74 million based on the projected annual costs shown in Fig 4.

### Epidemiological transition metrics

In our model, the IPR in 2030 would be 0.02 when scaling to 95-95-95 in all population groups, compared with 0.04 for the baseline scenario and the IMR would be >1 in both population groups.

## Discussion

Our modeling demonstrated that without scaling up diagnosis and treatment rates, new HIV infections are likely to increase in Botswana over the next 10 years. Including migrants (around 7% of the total population) who are mainly from Zimbabwe in the scale-up of services to reach 95-95-95 almost doubled the reduction in 2030 incidence. However, even if 95-95-95 was achieved in citizen and migrants, reductions in new infections would still fall well short of the 90% target outlined in the Fast-Track targets [30]. This implies that without a significant increase in the coverage of additional prevention measures, scaling up testing and treatment alone to 95-95-95 in all population groups will not meet HIV incidence reduction targets. While including migrants had the greatest benefits for the migrant population, the model demonstrated that meaningful benefits were also accrued among citizens in Botswana.

Despite some gains in reducing new HIV infections, our models suggest more is needed beyond meeting the 90-90-90 and 95-95-95 targets if the HIV incidence reduction targets are to be met in Botswana. Whilst our results for the projected care cascade are similar those recently reported by Kibona and Yang (2018), unlike us, they estimated that Botswana would meet the HIV incidence and HIV-related mortality targets by 2030 [31]. An important difference between the two models is that Kibona and Yang did not consider international migration, whether documented or undocumented. Additionally, Kibona and Yang calibrated their model with epidemiological data, including HIV incidence between from 2010 to 2016, but Botswana recorded an increase in new infections in 2017 (compared with drops in incidence in previous years). We used HIV incidence from 2017 in our model.

Botswana mostly over 60% of its HIV program and there is concern that expanding the programs to migrants would be an added cost to the country. However, expanding the program to include migrants would be cost-effective in the long run due to the reduction in new HIV

**Table 2. HIV infections, incidence, deaths, and PLHIV for various scenarios, 2020–30.**

| | | Baseline | | | | 95-95-95 Citizens only | | | | | 95-95-95 Citizens and Migrants | | | | | |
|---|---|---|---|---|---|---|---|---|---|---|---|---|---|---|---|---|
| | | In 2020 | In 2030 | 20–30 inclusive | 2030–2020% change | In 2020 | In 2030 | 20–30 inclusive | Averted (cpr baseline) | 2030–2020% change | In 2020 | In 2030 | 20–30 inclusive | Averted (cpr baseline) | Averted (cpr Cit ONLY) | 2030–2020% change |
| New HIV infections | Total | 13,015 | 18,813 | 171656 | 31% | 12,683 | 9,766 | 123,342 | 48,314 | -23% | 12,548 | 7,287 | 107,162 | 64,494 | 16,180 | -42% |
| | Citizen | 12,306 | 17,817 | 162,439 | 31% | 12,002 | 9,245 | 116,767 | 45,672 | -23% | 11,866 | 6,886 | 101,319 | 61,120 | 15,448 | -42% |
| | Mig | 709 | 996 | 9,216 | 29% | 681 | 521 | 6575 | 2,641 | -23% | 682 | 401 | 5844 | 3,372 | 731 | -41% |
| HIV incidence (per 100 p. y.) | Total | 0.67 | 0.61 | n/a | -9% | 0.65 | 0.31 | n/a | n/a | -52% | 0.64 | 0.23 | n/a | n/a | n/a | -64% |
| | Citizen | 0.68 | 0.62 | n/a | -9% | 0.67 | 0.32 | n/a | n/a | -52% | 0.66 | 0.24 | n/a | n/a | n/a | -64% |
| | Mig | 0.47 | 0.42 | n/a | -11% | 0.45 | 0.22 | n/a | n/a | -51% | 0.45 | 0.17 | n/a | n/a | n/a | -62% |
| HIV-related deaths | Total | 725 | 852 | 8,396 | 15% | 717 | 515 | 6,670 | 1,726 | -28% | 716 | 435 | 6,228 | 2,168 | 442 | -39% |
| | Citizen | 644 | 759 | 7,464 | 15% | 639 | 444 | 5,861 | 1,603 | -31% | 637 | 402 | 5,613 | 1,851 | 248 | -37% |
| | Mig | 80 | 92 | 928 | 13% | 78 | 71 | 809 | 119 | -9% | 79 | 33 | 616 | 312 | 193 | -58% |
| PLHIV | Total | 358,875 | 473,085 | n/a | 24% | 358,761 | 432,814 | n/a | n/a | 17% | 358,718 | 418,909 | n/a | n/a | n/a | 14% |
| | Citizen | 337,776 | 446,837 | n/a | 24% | 337,673 | 408,740 | n/a | n/a | 17% | 337,628 | 395,309 | n/a | n/a | n/a | 15% |
| | Mig | 21,098 | 26,248 | n/a | 20% | 21,088 | 24,073 | n/a | n/a | 12% | 21,090 | 23,600 | n/a | n/a | n/a | 11% |

Mig: migrant; PLHIV: people living with HIV; cpr: compared

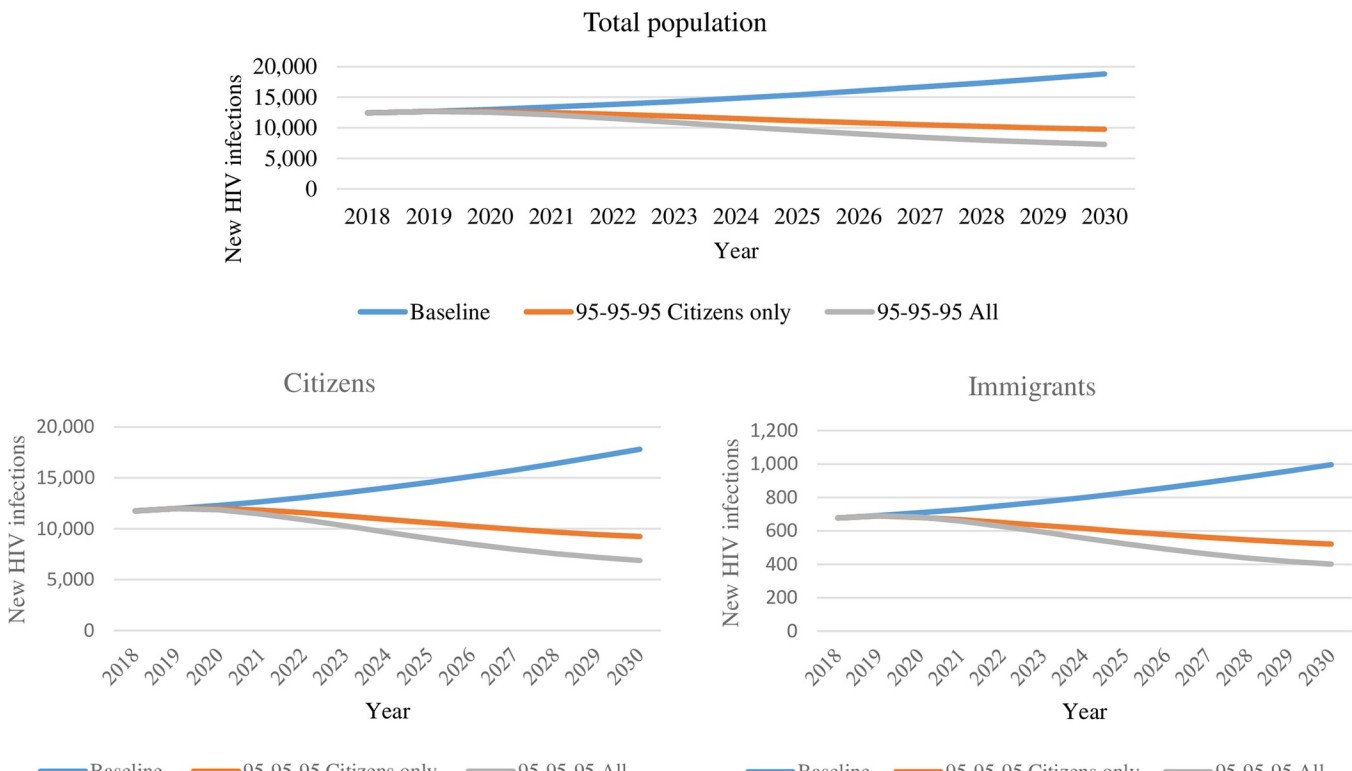

**Fig 2. New HIV infections.** Projected new HIV infections in the total population, citizens, and immigrants per different scenarios: baseline, 95-95-95 in citizens only, and 95-95-95 in all population groups.

infections. Williams et al. (2016) estimated that between 2016 and 2030, Botswana would reduce the total cost of the HIV program by USD26 million when expanding to reach the Fast-Track targets [29]. The Williams et al. model built on the high coverage and viral suppression reported by Gaolathe and colleagues (2016) and considered expanding ART coverage, male circumcision, and PrEP, but did not consider migrants [20]. Again the model and costing estimates did not call out migrants, similar to previous cost estimates for Botswana[32]. We estimated that treating migrants would cost USD74 million between 2020 and 2030. Further research is needed to examine the cost-effectiveness of treating migrants and the budgetary implications for Botswana.

By underscoring the considerable challenge associated with achieving global HIV incidence reduction targets, our findings are consistent with those of Scott et al. (2018), who modeled the Australian HIV epidemic. Modeling a best-case HIV treatment and care scenario among men who have sex with men (MSM) in Australia, they found that even if high-risk MSM had four HIV tests per year, ART, viral suppression, and PrEP, and condom use coverage was 100%, only an 80% reduction in HIV incidence could be achieved by 2030 [33]. While these findings apply to a highly concentrated HIV epidemic in Australia, Williams et al. (2017) obtained similar results when modeling a generalized epidemic [34]. Williams et al. found that only through a universally expanded program that included both prevention and treatment in South Africa can HIV incidence and mortality targets be met by 2030. Other models focusing on India and the USA have also failed to demonstrate 90% HIV incidence reduction by 2030 if Fast-Track targets are met [35, 36]. Our finding are also consistent with analysis of population-based surveys, surveillance, and routine program data used in Ethiopia that demonstrated that for HIV testing, ART, viral suppression, and AIDS-related deaths Fast-Track targets will be reached,

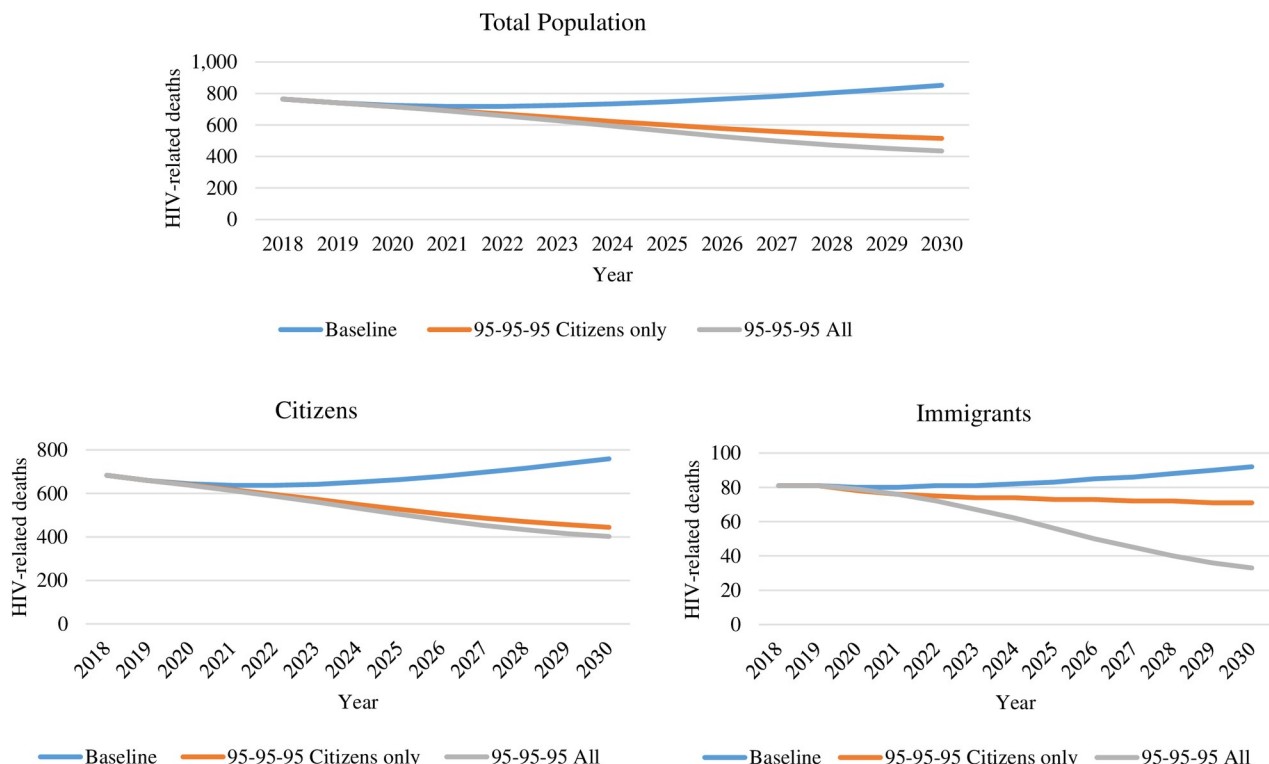

**Fig 3. HIV-related deaths.** Projected new HIV-related deaths in the total population, citizens, and immigrants per different scenarios: baseline, 95-95-95 in citizens only, and 95-95-95 in all population groups.

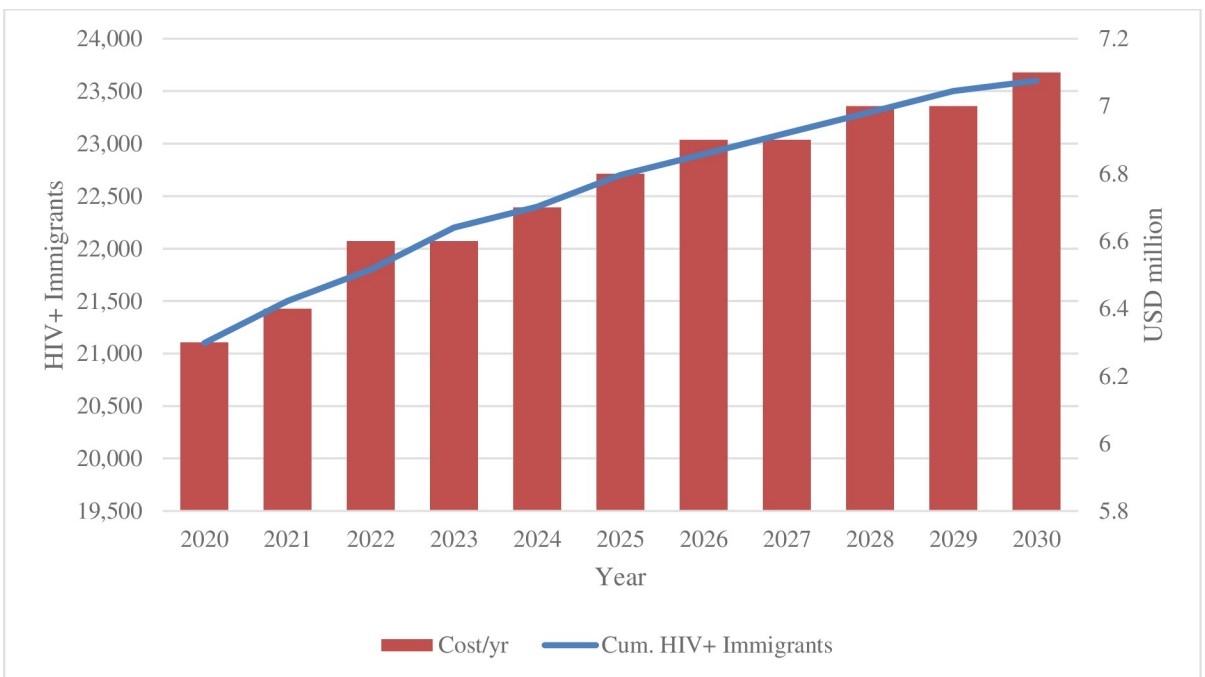

**Fig 4. Migrants living with HIV and cost of treatment per year.** Annual cost of treating the projected number of immigrants between 2020 and 2030

but not targets for reductions in new HIV infections based on progress made between 2011 and 2016 [37].

The Fast-Track targets are focused on the need to scale up HIV testing, linkage to care, ART initiation, adherence to ART, retention in care, and viral suppression. However, given the difficulty in achieving the 90% reduction in incidence even if the 95-95-95 targets are met, there is a need to focus on two key things. First that important subgroups in the population at risk of HIV infection, which includes migrants, are not excluded from testing, treatment and care. Second it highlights the central role of HIV prevention when trying to reduce HIV incidence. Various other factors are influencing this outcome including a lack of focus on reducing transmission among MSM and sex workers, and low uptake of HIV testing and prevention among young people [38].

It is important to monitor social and structural barriers to the HIV response in country programs in order to maintain focused strategies for HIV prevention and treatment. In Botswana, multiple concurrent partners, alcohol consumption, drug dependence, transactional sex, intergenerational sex, inconsistent condom use, high population mobility, and stigma and discrimination have been identified as barriers to HIV prevention [38]. Targeting these elements of the HIV epidemic in Botswana, including universal access to prevention and treatment for all population groups, will be required for Botswana to reach Fast-Track targets by 2030. In particular, targeting mobile men has been shown to significantly reduce HIV transmission in home communities, providing additional support for extending HIV treatment and care to all populations [39].

The use of additional epidemiological transition metrics for tracking the HIV epidemic, such as the incidence:prevalence ratio (IPR) and incidence:mortality ratio (IMR), have been proposed and used in UNAIDS reports [2, 6, 40]. IPR is ratio of the number of new infections to the number of PLHIV; an IPR at or below the benchmark of 0.03 signals that progress is on track to end AIDS as a public health threat. In our model, the IPR would signify significant progress while scaling to 95-95-95 in all population groups. The IPR typically reduces when incidence is lower than all-cause mortality among PLHIV. IMR is the ratio of the number of new HIV infections to all-cause mortality. In our model, the IMR was >1, in line with a growing population of PLHIV.

Our study has several limitations. The model required a burn-in period of 10 years prior to 2020 to stabilize and replicate Botswana's epidemic trends. The reliability of estimated changes in HIV incidence and mortality under various scenarios may have been affected by unknown data and associated assumptions. Age transitions were not included in the model, meaning that we did not factor in young people as they grew into the modelled age group (16–65 years). The model also made simplifying assumptions about the migrant population due to the absence of data: we did not account for dynamic migration and imported infections among migrants (e.g. the migrant population only accounted for an initial prevalence, a net change in population size over time, and new infections occurring within Botswana), and migrants were assumed to mix randomly with non-migrants (if assortativity among migrants was higher, this would reduce the impact that treating migrants has for non-migrants compared to what the model has projected). As with other models, our analysis was dependent on the availability, reliability, and quality of data inputs. Trends in new infections have varied in Botswana; in particular, the number of new infections increased in 2017. Large fluctuations reduce the model's reliability. More generally, mathematical models are simplifications of reality and cannot capture all aspects of the real world, hence it is important to understand the assumptions and parameters used in the model. We believe that while there is uncertainty in the model inputs where assumptions were made due to missing data, the projected trends and relative differences are reasonable estimates that reflect the best available data at the time of modeling.

## Conclusion

The inclusion of migrants in Botswana's national HIV care and treatment program would result in important prevention and health benefits for both citizens and migrants. Including migrants in Botswana's mainstream HIV care and treatment program was estimated to avert an additional 16,000 new HIV infections, a 34% increase. However, while inclusion of migrants will accelerate the rate of reduction of new HIV infections by 2030, this will not be sufficient to reach the 95-95-95 incident reduction target. More efficiencies and higher coverage levels are needed, utilizing current prevention modalities (e.g. condom use, voluntary medical male circumcision, treatment as prevention, and PrEP) and ensuring equitable access to treatment in all populations groups in order to meet the 2030 target for HIV incidence reduction [41].

## Supporting information

**S1 Table. Botswana data inputs.** Full data inputs for the model with justifications where applicable.
(DOCX)

**S2 Table. Model fitting parameters.** Full set of parameters used to fit the model.
(DOCX)

**S3 Table. Baseline scenario results (2010–30).** Table shows the full results for the baseline scenario. Variables include: new HIV infections, HIV related deaths, number of people living with HIV (PLHIV), prevalence and incidence for immigrants and citizens.
(DOCX)

**S4 Table. Scaling up to 95-95-95 in citizens only.** Table shows the full results for the scenario when scaling up in citizens only. Variables include: new HIV infections, HIV related deaths, number of people living with HIV (PLHIV), prevalence and incidence for immigrants and citizens.
(DOCX)

**S5 Table. Scaling up to 95-95-95 in both citizens and migrants.** Table shows the full results for the scenario when scaling up in both immigrants and citizens. Variables include: new HIV infections, HIV related deaths, number of people living with HIV (PLHIV), prevalence and incidence for immigrants and citizens.
(DOCX)

**S1 Fig. Calibration of the model and the fit achieved.** Figure shows alongside model fitting parameters the fit that was achieved for the model.
(DOCX)

## Acknowledgments

Mark Minnery and Maria del Mar Quiroga for helping with Optima model logistics. Campbell Aitken for editing and proof reading the manuscript.

## Author Contributions

**Conceptualization:** Tafireyi Marukutira, Nick Scott, Sherrie L. Kelly, Charles Birungi, Suzanne Crowe, Mark Stoove, Margaret Hellard.

**Data curation:** Tafireyi Marukutira, Nick Scott.

**Formal analysis:** Tafireyi Marukutira, Nick Scott.

**Methodology:** Tafireyi Marukutira, Nick Scott, Sherrie L. Kelly, Charles Birungi, Joseph M. Makhema.

**Project administration:** Tafireyi Marukutira, Nick Scott.

**Software:** Nick Scott.

**Supervision:** Nick Scott, Sherrie L. Kelly, Suzanne Crowe, Mark Stoove, Margaret Hellard.

**Visualization:** Nick Scott.

**Writing – original draft:** Tafireyi Marukutira, Nick Scott.

**Writing – review & editing:** Tafireyi Marukutira, Nick Scott, Sherrie L. Kelly, Charles Birungi, Joseph M. Makhema, Suzanne Crowe, Mark Stoove, Margaret Hellard.

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
