## [Decision Letter · Decision Letter 0]

9 Oct 2019

PONE-D-19-24757

Modelling the impact of migrants on the success of the HIV care and treatment program in Botswana

PLOS ONE

Dear Dr. Marukutira,

Thank you for submitting your manuscript to PLOS ONE. After careful consideration, we feel that it has merit but does not fully meet PLOS ONE’s publication criteria as it currently stands. Therefore, we invite you to submit a revised version of the manuscript that addresses the points raised during the review process.

We would appreciate receiving your revised manuscript by Nov 23 2019 11:59PM. To enhance the reproducibility of your results, we recommend that if applicable you deposit your laboratory protocols in protocols.io, where a protocol can be assigned its own identifier (DOI) such that it can be cited independently in the future. For instructions see: http://journals.plos.org/plosone/s/submission-guidelines#loc-laboratory-protocols

We look forward to receiving your revised manuscript.

Kind regards,

Justyna Dominika Kowalska

Academic Editor

PLOS ONE

Journal Requirements:

Additional Editor Comments (if provided):

Reviewers' comments:

Reviewer's Responses to Questions

**Comments to the Author**

1. Is the manuscript technically sound, and do the data support the conclusions?

Reviewer #1: Yes

Reviewer #2: Yes

2. Has the statistical analysis been performed appropriately and rigorously? 

Reviewer #1: Yes

Reviewer #2: N/A

3. Have the authors made all data underlying the findings in their manuscript fully available?

Reviewer #1: Yes

Reviewer #2: Yes

4. Is the manuscript presented in an intelligible fashion and written in standard English?

Reviewer #1: Yes

Reviewer #2: Yes

5. Review Comments to the Author

Reviewer #1: The paper is an interesting addition to the growing body of research addressing challenges to reach the UNAIDS goals of radical reduction of incidence and mortality by 2030, even in case of achieving 90-90-90 by 2020 and 95-95-95 by 2030.

The hypothesis put forward in the manuscript is that the suboptimal reduction in incidence is due the fact that immigrants in Botswana are not eligible for the free national HIV care and treatment. This hypothesis is investigated through a modelling study using the well established Optima HIV model.

While overall the manuscript is very clear, additional explanation of some key aspects would help the reader to better understand the situation and the modelling approach:

- Do the data sources used for migrants represent only documented migrants (or both documented and undocumented)? It should be explained in the methods.

- Is the migrant population stable or dynamic (new migrants arriving, some emigrating) and is this accounted for by the model? In particular am I understanding correctly that only new infections in Botswana are considered and not the ones that are imported? Is this a limitation?

- The Table 1 and Appendix 1 provide data inputs for the model. Some of them I presume were not available or were not available directly and some values had to be assumed. It would be good to explain (perhaps in the Appendix) where these values come from and why different values are assumed for migrants and for the citizens (e.g. in case of time to be linked to care or in treatment failure rates.

- Further in the Table 1 some values are provided without the reference year – does this mean that this number was applied to all years in the considered time period?

- I would suggest to expand the explanation of the model calibration. Even if this follows a previously published procedure it would help the reader to understand the results. Specifically, the description sais that the parameters for the force of infection were adjusted, but the way they were adjusted is not explained. Given the uniform effect of scaling up the treatment coverage both on the citizens and on the migrants I believe that random mixing of these two population subgroups is assumed in the model? If this is the case I would suggest to explain whether or not this assumption is likely.

- The discussion mentions additional epidemiological measures: IPR and IMR. For clarity it would be better to describe these in the results. Alternatively to drop this paragraph from discussion.

Reviewer #2: Manuscript "Modelling the impact of migrants on the success of the HIV care and treatment program in Botswana" is technically sound and the data support the conclusion. The English language is clear and easy to follow and each section (introduction, methods... etc) contains the necessary elements.

I have editorial comments:

1. The citation should be in the end of the sentence (citations [6], [7-9], [21,22], [41] do not meet these criteria)

2. Table 1 - Tables should not have lines on the left end and on the right end and there should be no lines crisscrossing it. Aline on top and a line below is all that makes the work neater.

Regarding the epidemiological aspect - in my opinion it would be worth to know from where mainly the migrants in Botswana come from? Besides, the manuscript is valuable, includes the logic way of thinking and draws attention to the fact that the migration should be taken into consideration in public health programs. This is important information from the clinical point of view.

6. PLOS authors have the option to publish the peer review history of their article (what does this mean?). If published, this will include your full peer review and any attached files.

Reviewer #1: No

Reviewer #2: No

---

## [Author Response · Author response to Decision Letter 0]

17 Oct 2019

Responses to reviewers

Reviewer #1: The paper is an interesting addition to the growing body of research addressing challenges to reach the UNAIDS goals of radical reduction of incidence and mortality by 2030, even in case of achieving 90-90-90 by 2020 and 95-95-95 by 2030.

The hypothesis put forward in the manuscript is that the suboptimal reduction in incidence is due the fact that immigrants in Botswana are not eligible for the free national HIV care and treatment. This hypothesis is investigated through a modelling study using the well-established Optima HIV model.

While overall the manuscript is very clear, additional explanation of some key aspects would help the reader to better understand the situation and the modelling approach:

1. Do the data sources used for migrants represent only documented migrants (or both documented and undocumented)? It should be explained in the methods.

Response: The data includes documented migrants only. This has been updated in the introduction and methods sections.

2. Is the migrant population stable or dynamic (new migrants arriving, some emigrating) and is this accounted for by the model? In particular, am I understanding correctly that only new infections in Botswana are considered and not the ones that are imported? Is this a limitation?

Response: The migrant population only accounted for an initial prevalence, a net change in population size over time, and new infections occurring within Botswana. These points have been added as limitations.

3. The Table 1 and Appendix 1 provide data inputs for the model. Some of them I presume were not available or were not available directly and some values had to be assumed. It would be good to explain (perhaps in the Appendix) where these values come from and why different values are assumed for migrants and for the citizens (e.g. in case of time to be linked to care or in treatment failure rates.

Response: Unavailable data inputs were assumed and are all marked “*”. Explanations of sources and differences between migrants and citizens have been added in Appendix 1. 

4. Further in the Table 1 some values are provided without the reference year – does this mean that this number was applied to all years in the considered time period?

Response: Assumptions were made for 2017 as data inputs in the model. Table 1 and Appendix 1 have been updated to reflect this.

5. I would suggest to expand the explanation of the model calibration. Even if this follows a previously published procedure it would help the reader to understand the results. Specifically, the description says that the parameters for the force of infection were adjusted, but the way they were adjusted is not explained. Given the uniform effect of scaling up the treatment coverage both on the citizens and on the migrants I believe that random mixing of these two population subgroups is assumed in the model? If this is the case, I would suggest to explain whether or not this assumption is likely.

Response: We have provided additional information regarding the calibration procedure. 

It is correct that the model assumed random mixing between the population groups. It is unclear whether this is likely or not, as no data is available, and we have added this to the limitations.

6. The discussion mentions additional epidemiological measures: IPR and IMR. For clarity it would be better to describe these in the results. Alternatively, to drop this paragraph from discussion.

Response: Epidemiological transition measures have been discussed at UNAIDS and are increasingly becoming relevant as countries track progress in the HIV epidemic response. While some other metrics have been discussed, IPR and IMR are currently gaining momentum as markers of progress. Therefore, we have added in the projected IPR/IMR in 2030 in the results section: 

“Epidemiological transition metrics

In our model, the IPR in 2030 would be 0.02 when scaling to 95-95-95 in all population groups, compared with 0.04 for the baseline scenario and the IMR would be >1 in both population groups.” 

Reviewer #2: Manuscript "Modelling the impact of migrants on the success of the HIV care and treatment program in Botswana" is technically sound and the data support the conclusion. The English language is clear and easy to follow and each section (introduction, methods... etc) contains the necessary elements.

I have editorial comments:

7. The citation should be in the end of the sentence (citations [6], [7-9], [21,22], [41] do not meet these criteria)

Response: Thank you for picking this up. Corrections have been made on all citations that were not at the end of a sentence. 

8. Table 1 - Tables should not have lines on the left end and on the right end and there should be no lines crisscrossing it. Aline on top and a line below is all that makes the work neater.

Response: We had used the table format as shown on the journal page’s instruction to authors and manuscript example. Based on the reviewer’s comment, Table 1 and 2 have been updated to follow the suggested format.

9. Regarding the epidemiological aspect - in my opinion it would be worth to know from where mainly the migrants in Botswana come from? Besides, the manuscript is valuable, includes the logic way of thinking and draws attention to the fact that the migration should be taken into consideration in public health programs. This is important information from the clinical point of view.

Response: The origin of most migrants to Botswana has been added in the background and discussion (mainly Zimbabwe).

---

## [Decision Letter · Decision Letter 1]

27 Nov 2019

Modelling the impact of migrants on the success of the HIV care and treatment program in Botswana

PONE-D-19-24757R1

Dear Dr. Marukutira,

We are pleased to inform you that your manuscript has been judged scientifically suitable for publication and will be formally accepted for publication once it complies with all outstanding technical requirements.

With kind regards,

Justyna Dominika Kowalska

Academic Editor

PLOS ONE

Additional Editor Comments (optional):

Reviewers' comments:

Reviewer's Responses to Questions

**Comments to the Author**

1. If the authors have adequately addressed your comments raised in a previous round of review and you feel that this manuscript is now acceptable for publication, you may indicate that here to bypass the “Comments to the Author” section, enter your conflict of interest statement in the “Confidential to Editor” section, and submit your "Accept" recommendation.

Reviewer #1: All comments have been addressed

2. Is the manuscript technically sound, and do the data support the conclusions?

Reviewer #1: Yes

3. Has the statistical analysis been performed appropriately and rigorously? 

Reviewer #1: Yes

4. Have the authors made all data underlying the findings in their manuscript fully available?

Reviewer #1: Yes

5. Is the manuscript presented in an intelligible fashion and written in standard English?

Reviewer #1: Yes

6. Review Comments to the Author

Reviewer #1: The abbreviations for the epidemic transition parameters (IPR and IMR) should be explained in the text in the first place where they are used (i.e. the new paragraph in the Results secion).

7. PLOS authors have the option to publish the peer review history of their article (what does this mean?). If published, this will include your full peer review and any attached files.

Reviewer #1: No

---

## [Editor Report · Acceptance letter]

13 Dec 2019

PONE-D-19-24757R1 

Modelling the impact of migrants on the success of the HIV care and treatment program in Botswana 

Dear Dr. Marukutira:

I am pleased to inform you that your manuscript has been deemed suitable for publication in PLOS ONE. Congratulations! Your manuscript is now with our production department. 

With kind regards,

on behalf of

Dr. Justyna Dominika Kowalska 

Academic Editor

PLOS ONE